# Higher Education Students' Perceived Readiness for Computer-Supported Collaborative Learning

**Ghodratolah Khalifeh [1], Omid Noroozi [2], Mohammadreza Farrokhnia [2,\*] and Ebrahim Talaee [3]**

[1]   Shahid Chamran University of Ahvaz, Ahvaz 6135783151, Iran; Ghodratkhalife@yahoo.com
[2]   Wageningen University and Research, 6706 KN Wageningen, The Netherlands; Omid.Noroozi@wur.nl
[3]   Tarbiat Modares University, Tehran 1411713116, Iran; E.Talaee@modares.ac.ir
\*   Correspondence: Mohammadreza.Farrokhnia@wur.nl

**Abstract:** The purpose of this research was to study the perceived readiness of higher education students for computer-supported collaborative learning (CSCL). Moreover, the role of important demographic variables, such as gender, major of study, and computer ownership, was examined in students' perceived readiness and its sub-scales. The data was collected from 326 higher education students of four study groups from a state university in Iran. MANOVA analysis was conducted to explore the possible role of the demographic variables in students' perceived readiness for CSCL. Most of the participants showed high readiness for CSCL. The male participants demonstrated more online learning aptitude compared to females. A statistically significant difference was found in the online learning aptitude of the respondents majoring in engineering and basic sciences with the rest of the participants. Furthermore, the students with a personal computer, laptop, or tablet demonstrated higher levels of readiness for CSCL and online learning aptitude.

**Keywords:** computer-supported collaborative learning (CSCL); perceived readiness; technology readiness; higher education students; gender; major of study; computer ownership

---

## 1. Introduction

The rapidly increasing use of computers in education, and especially the migration of many university courses to web-based delivery, has triggered a resurgence of interest among educators in non-traditional methods of course design and delivery [1]. Over the past two decades, computer-supported collaborative learning (CSCL) environments have received attention in various educational settings, especially in higher education [2,3]. CSCL is built on the premise that collaborative learning and knowledge construction can effectively be supported by technology [4–6].

CSCL environments can promote students' content understanding and domain-specific knowledge acquisition [6–11], contribute to students' motivation [12,13], foster development of higher-order thinking skills and metacognitive skills [14], and enhance the development of prosocial behavior, such as showing empathy and helping others [15].

However, scholars have reported that not all students could harness the potentials of CSCL environments [16,17]. Indeed, in contrast to the positive findings, there is also a vast body of research literature reporting mixed or negative findings [18,19]. Scholars assert that given the nature of CSCL (i.e., technological and collaborative dependence, high level of agency, and autonomy), an inadequate level of students' readiness for CSCL is one of the important reasons for such discrepancy in terms of mixed positive and negative findings [20,21].

Conley [22] defined the student's readiness as a multi-faceted variable that included factors that were both internal and external to the learning environment. Such collaborative, autonomous,

and technological nature of CSCL environments implies that proper readiness of students is needed for active and effective engagement in CSCL environments. In this sense, Xiong et al. [21] argued that students' psychological readiness, collaborative learning skills, and technology readiness for online learning environments were the main factors that might have a role in their readiness for CSCL. Furthermore, there are different demographic variables, such as age, gender, the major of the study, and computer ownership [23–25], that may have a role in how students will be prepared to work effectively in CSCL environments. So far, most studies have focused on exploring the role of these factors (in isolation and not in combination) for students' readiness for learning in online environments in general and not in CSCL environments. Therefore, this study aimed to empirically investigate the role of demographic variables in students' readiness for learning in CSCL environments.

### 1.1. Students' Psychological Readiness for CSCL

Psychological readiness focuses on an individual's state of mind as this impacts on the outcome of any kind of computer-based learning environments [26]. Among different psychological factors, motivation to engage in online discussions has been found to be one of the important factors that significantly play a role in students' active participation in CSCL environments [15,27,28]. As motivation is a multidimensional and multilevel construct [29], a wide variety of definitions are discussed and used in educational psychology research. We adopted the concept of motivation developed by Deci and Ryan [30], where "[t]o be motivated means to be moved to do something". According to this definition, a person who feels impetus or inspiration to act is thus characterized as motivated [31]. In self-determination theory, Ryan and Deci [31] distinguished between different types of motivation based on the various goals that give rise to an action, namely intrinsic and extrinsic motivation. While intrinsically motivated learning is highly dependent on the learners' interest, self-efficacy, and perceived value of the activity of learning itself, the extrinsically motivated learning is contingent on external stimuli, such as reinforcement, avoiding punishment, or complying with social norms [21,32].

Although some scholars stated that generally, strong motivation is a prerequisite for CSCL environments [33–35], being only extrinsically motivated might result in low-level participation and cognitive engagement in CSCL environments [36]. In this regard, studies have reported that students with high intrinsic motivation demonstrate greater persistence [37], better ability to cope with failure [38], and higher-quality task engagement [31] in CSCL.

### 1.2. Students' Skills for Collaborative Learning

For active learning in the CSCL environment, students need to have different inter- and intrapersonal skills [39–42]. While interpersonal skills are required to overcome interpersonal conflict and to maintain positive interpersonal relationships with other members within a group [16,43], intrapersonal skills are necessary for students to manage their learning trajectories through the notion of independence and self-directedness in learning [44]. Regarding interpersonal skills, scholars argue that collaborative learning will not be productive if group members lack the interpersonal skills needed to cooperate effectively [45,46]. In this regard, skills, such as communication, resolving conflicts, and problem-solving, are among the most important interpersonal skills that students need to take the initiative in collaborative learning environments and become constructive in the ways they deal with conflict [21,47,48].

Furthermore, scholars assert that collaborative learning environments demand learners to regulate their learning through shared metacognitive monitoring and control of motivation, cognition, and behavior [49]. Thus, these environments require learners whom Zimmerman [50] refers to as self-regulated. Self-regulated learners are usually focused on regulating their own knowledge and behavior, with no intention of influencing other students [51]. Therefore, it is considered an intrapersonal skill [52]. Self-regulated learners engage in self-observation, self-judgment, self-reactions, and self-management, which can go simultaneously [53]. Based on the previous literature, among different aspects, self-management is considered one of the most important intrapersonal

skills [21,54] that promote self-regulated learning in CSCL environments [55]. Self-management refers to students' ability to motive themselves, monitor their understanding, work autonomously, collaborate with others, and have the desire to self-improve by acquiring new knowledge and skills [56].

### 1.3. Students' Technology Readiness

Parasuraman [57] defined technology readiness as one's propensity to embrace new technologies for accomplishing goals in life, work, or educational contexts. It is a combination of positive and negative technology-related beliefs that are assumed to vary among individuals [58]. Scholars argue that these coexisting beliefs determine the predisposition of an individual to engage with new technology [59]. In this regard, technology self-efficacy is considered the most important factor that may influence the students' technology readiness. Generally, self-efficacy is defined as one's personal judgment of his or her ability to succeed in the performance of a particular task or skill [60]. However, with the prevalence of technology in educational contexts, different kinds of technology self-efficacy, e.g., computer, internet, and information technology self-efficacy, have been the focus of many scholars [61–64]. In a broad definition, technology self-efficacy refers to the beliefs, values, confidence, and comfort that an individual has while using technology for different purposes [65].

Besides, scholars assert other influential factors, such as computer ownership, major of study, and gender, can also have a role in students' technological readiness [23] by triggering their motivation, technical skills, and technology acceptance [66–68].

### 1.3.1. Computer Ownership

Technology access is among the important factors that may play a role in students' technology self-efficacy and readiness [69]. Here, technology access is related to the availability of technological devices, such as computers, laptops, or tablets, not only at school but also at home. Previous studies have provided evidence that technology access is the key predictor for students' technology readiness in online learning environments [70,71]. According to Woodrow [72], technology access in the form of computer ownership is a confidence-building factor that can help in mitigating fear and anxiety about using computers. In the same vein, Ibrahim et al.'s [73] findings revealed the individuals who used computers more often (with an average of 22.5 h per week) had a higher level of readiness for online learning.

Furthermore, according to Nami and Vaezi [23], computer ownership has a significant role in the acquisition of technological knowledge and perceived ease of use. As it was shown in McCoy's [74] study, participants, who owned a computer, scored significantly better in technology knowledge scale compared to the non-owners. The results of Basol, Cigdem, and Unver's [75] study also indicated that previous computer use and ownership were significant predictors of students' readiness for online learning.

### 1.3.2. Major of Study

Ishtaiwa [76], in his research with university students at Al Ain University of Science and Technology in UAE, showed that students' academic major significantly played a role in their use and perceived effects of technology for learning in online settings. In the same line, Kumar and Mahajan [77] revealed significant differences in computer and internet use among various academic majors in the Indian higher education setting. In their study with different Iranian university students, Nami and Vaezi [23] observed that academic major was a significant factor in technology knowledge and skills as computer engineering students scored better on these scales. This finding was in line with the result of Sun and Rueda's [24] study in the Schools of Gerontology and Engineering at a large research university in the south-western USA, who observed higher computer self-efficacy among students in the school of engineering.

### 1.3.3. Gender Differences

The results of previous researches that explore the role of gender differences in students' readiness and acceptance of online learning environments are contradictory. Although some studies have confirmed that there are differences between men and women in terms of online learning behavior, there is still no consensus among scholars in this regard. For instance, some studies have reported that female students outperform male students in online learning environments [78,79] with a higher sense of community [80] and a higher degree of online activity and discussion [81]. However, other studies have shown that the perceptions towards computer self-efficacy, perceived usefulness, ease of use, and behavioral intention to use e-learning have been higher for men than those of women [71–73]. Also, in contradicting these findings, other scholars have pointed out that there are no differences among males and females regarding their engagement in online learning, achievement, motivation, and satisfaction [82–84].

Regarding the inconsistencies mentioned above, scholars argue that cultural and contextual differences play an important role [82,85]. Nai and Kirkup [85], in their study with Chinese and British students, reported that gender differences in the use of and attitudes toward the internet and computers were higher in the British group than the Chinese group. In the other study, Hannon and D'Netto [86] confirmed that cultural differences had a significant role in student engagement with learning technologies.

As mentioned above, many factors contribute to students' readiness for the CSCL environment. Thus, "before-collaboration" evaluation of students' readiness can provide valuable information that may help instructors to cultivate students' collaborative learning readiness through re-thinking about group configuration or following up actions, which need to be applied to augment their readiness levels. However, to date, most of the previous empirical research on CSCL has focused on the process of collaboration and the outcome of online learning environments, as after- and during- collaboration assessments [87].

Thus, given the high demand for developing CSCL environments for higher education programs worldwide, it is worth assessing students' readiness before developing any technological learning environment. The results can further help course designers and policymakers to consider better the needs and capabilities of students in designing the learning environments. Furthermore, although many studies have explored the role of other influential factors, i.e., computer ownership, major of study, and gender, in students' readiness for the online learning environments, no study has examined the role that these factors play in students' readiness for the CSCL environments. As a result, this study seeks to answer the following questions:

**Research Question 1.** To what extent are the higher education students ready for learning in CSCL environments?

**Research Question 2.** What are the roles of students' computer ownership, major of study, and gender in their readiness for learning in CSCL environments?

## 2. Materials and Methods

### 2.1. Participants

A convenient sample of undergraduate freshmen students (N = 350) from four groups of humanities, engineering, basic sciences, and agricultural sciences from a state university in Ahvaz, Iran, participated in this study. The participants included 125 males and 201 females with the age range of 18 to 24 years. Although most of the sample population consisted of women, these participants reflected the gender distribution of undergraduate students at the university under study.

## 2.2. Materials

Students' Readiness for CSCL

The Xiong et al.'s [21] instrument on students' readiness for CSCL was utilized to measure students' perceived readiness for CSCL environments. The instrument includes 39 items divided into three sub-scales, i.e., motivation for collaborative learning, prospective behaviors for collaborative learning, and online learning aptitude (See Table 1). The items are rated on a 5-point Likert scale, ranging from "strongly disagree" (=1) to "strongly agree" (=5). Furthermore, students' demographic information was gathered through a survey, including age, gender, major, and computer ownership, based on participant's self-report.

**Table 1.** Students' readiness for CSCL instrument adopted from Xiong et al. [21].

| Sub-Scales | Aspects | Sample |
|---|---|---|
| Motivation | Interest Perceived value Self-efficacy Reinforcement | The possible reason I would like to participate in collaborative learning is, it is fun it can help my academic learning I believe I can work well with my groupmates I hope to have a good relationship with my groupmates |
| Prospective behaviors | Communication Conflict-resolution Problem-solving Self-management | If I am doing group work, I would listen to other members' ideas I would be able to implement an appropriate conflict resolution strategy I would exercise appropriate participation accordingly I would be able to monitor my group's progress |
| Online learning aptitude | Technical skills Comfort | I am good at using the internet to communicate with others effectively. I am willing to use online communication tools to do group work with my groupmates. |

Also, the reliability of the students' readiness for the CSCL instrument was estimated in this study, obtaining a satisfactory coefficient for the whole instrument and its different sub-scales (see Table 2).

**Table 2.** The reliability of the instrument and its sub-scales.

| | Cronbach's Alpha | N of Items |
|---|---|---|
| The instrument | 0.94 | 39 |
| Motivation sub-scale | 0.90 | 15 |
| Prospective behaviors sub-scale | 0.90 | 15 |
| Online learning aptitude sub-scale | 0.85 | 9 |

Note. N = Number.

## 2.3. Procedure

Different variables included students' readiness for CSCL environments, and demographic information was collected in three 1.5-hour sessions. At the beginning of the sessions, the participants were fully informed about the main objectives of the research, and the participant's consent to participate in the research was obtained and recorded. As a part of the consent letter, the participants were given the right to fill in the questionnaire or not. As a result, of the 350 distributed questionnaires, 339 were completed, giving a response rate of 96%. After the deletion of responses with a high rate of missing data, 326 fully completed responses were retained for further analysis.

## 3. Analysis

*Statistical Tests*

To examine the role of gender, major of study, and computer ownership in students' readiness for CSCL and its sub-scales, a multivariate analysis of variance (MANOVA) was used. The main

objective of the MANOVA test was to examine whether or not the independent demographic variables (i.e., gender, major of study, and computer ownership) simultaneously explained statistically significant differences in the dependent variables (i.e., students' readiness for CSCL and its sub-scales).

## 4. Results

### 4.1. Descriptive Statistics

#### 4.1.1. Demographic Information

Table 3 summarizes the demographic information of the participants. As shown among the majors, in terms of the background of respondents, the humanities group had the highest percentage (52.45%), and the agricultural science group had the lowest percentage (7.67%). Also, out of the 326 participants, 76% owned computers, laptops, or tablets.

**Table 3.** Demographic characteristics of the participants.

|  |  | **N** | **Perc.** |
|---|---|---|---|
| Gender | Female | 201 | 61.66 |
|  | Male | 125 | 38.34 |
| Major of study | Humanities | 171 | 52.45 |
|  | Engineering | 85 | 26.08 |
|  | Basic sciences | 45 | 13.80 |
|  | Agricultural sciences | 25 | 7.67 |
| Computer ownership | Personal computer | 42 | 12.88 |
|  | Laptop | 142 | 43.56 |
|  | Tablet | 64 | 19.63 |

Note. N = Number. Perc. = Percentage.

#### 4.1.2. Students' Readiness for CSCL

Based on Nami and Vaezi [23], the frequency analysis and percentage of the respondents' scores in terms of their perceived readiness level for learning in CSCL environments were calculated and shown in Table 4. For each sub-scale, the minimum and maximum possible scores, i.e., score range, were related to the condition that all items were answered with 1 (i.e., strongly disagree) or 5 (i.e., strongly agree), respectively. According to Table 4, 86.81% of students perceived that they were ready for learning in CSCL environments. Also, based on their responses, a high percentage of students perceived to have a high motivation (86.20%), prospective behaviors (89.57%), and online learning aptitude (80.06%) for CSCL environments.

**Table 4.** Frequency and percentage of Likert scale responses.

| Scale | Sub-Scales | Likert Scale | Score Range | Freq. | Perc. |
|---|---|---|---|---|---|
| Students' readiness for CSCL | Motivation | Strongly disagree-disagree | 15–30 | 5 | 1.53 |
|  |  | Uncertain | 31–45 | 40 | 12.27 |
|  |  | Agree—strongly agree | 46–75 | 281 | 86.20 |
|  | Prospective behaviors | Strongly disagree-disagree | 15–30 | 6 | 1.84 |
|  |  | Uncertain | 31–45 | 28 | 8.59 |
|  |  | Agree—strongly agree | 46–75 | 292 | 89.57 |
|  | Online learning aptitude | Strongly disagree-disagree | 9–18 | 4 | 1.23 |
|  |  | Uncertain | 19–27 | 61 | 18.71 |
|  |  | Agree—strongly agree | 28–45 | 261 | 80.06 |
|  | Total | Strongly disagree-disagree | 39–78 | 2 | 0.61 |
|  |  | Uncertain | 79–117 | 41 | 12.58 |
|  |  | Agree—strongly agree | 118–195 | 283 | 86.81 |

Note. Freq. = Frequency. Perc. = Percentage.

*4.2. Research Question 1*

Based on the frequency analysis and percentage of the respondents' scores, the mean scores for each sub-scale were as follows: students' motivation (M = 55.90, SD = 9.26), prospective behaviors (M = 56.31, SD = 9.04), online learning aptitude (M = 33.02, SD = 6.22). Also, the mean score obtained in the students' readiness for CSCL (M = 145.23, SD = 21.10) indicated that the students had a relatively high level of readiness. Additionally, the means and standard deviations of the responses from the students' readiness for CSCL questions in which students were asked to answer on a Likert scale are presented in Table 5.

**Table 5.** Frequency analysis and percentage of the respondents' scores.

|  | N | Min. | Max. | M | SD |
|---|---|---|---|---|---|
| Motivation | 15 | 3.56 | 3.88 | 3.73 | 0.09 |
| Prospective behaviors | 15 | 3.61 | 3.93 | 3.74 | 0.09 |
| Online learning aptitude | 9 | 3.46 | 3.78 | 3.67 | 0.09 |
| Students' readiness for CSCL | 39 | 3.46 | 3.93 | 3.72 | 0.10 |

Note. N = Number. Min. = Minimum. Max. = Maximum. M = Mean. SD = Standard deviation.

*4.3. Research Question 2*

Research question 2 focused on the role of gender, major of study, and computer ownership in students' readiness for CSCL and its different sub-scales.

4.3.1. Gender

There was no significant difference between male and female respondents on students' readiness for CSCL (F = 3.12, p = 0.08) and two other dependent variables, i.e., motivation (F = 1.02, p = 0.31) and prospective behaviors for collaborative learning (F = 1.35, p = 0.25). However, online learning aptitude was significantly different between men and women (F = 7.90, p < 0.01). Comparing the mean scores also confirmed these observations. The male respondents had relatively higher mean scores (M = 34.24, SD = 6.54) than the females (M = 32.27, SD = 5.49) on the scale.

4.3.2. Major of Study

The major of the study did not play a role in motivation for collaborative learning, prospective behaviors for collaborative learning, and students' readiness for CSCL. The major of study only had a significant role in online learning aptitude (F = 4.14, *p < 0.01*). Based on Scheffe's posthoc test, engineering and basic sciences groups were different (*p < 0.05)* from the humanities group in terms of having higher online learning aptitude.

4.3.3. Computer Ownership

The results of MONOVA showed that owing electronic devices, such as a computer, laptop, or tablet, did not play a significant role in two dependent variables, i.e., motivation for collaborative learning and prospective behaviors for collaborative learning. However, owning device had a significant role in two other dependent variables, including students' perceived readiness for CSCL (F = 5.48, *p < 0.01*) and online learning aptitude (F = 18.12, *p < 0.01*).

**5. Discussion**

*5.1. Research Question 1*

The overall aim of this study was to assess the higher education students' readiness for CSCL. To achieve this goal, higher education students were evaluated based on three scales, i.e., motivation for collaborative learning, prospective behaviors for collaborative learning, and online learning aptitude.

The findings of this study showed that students' perceived readiness for CSCL was, generally, satisfactory. In line with previous studies, both intrinsic motivations and extrinsic motivations had a great influence on the motivation level of the students [31,36]. In the prospective behaviors for collaborative learning scale, both inter- and intrapersonal skills played an important role. This was consistent with the findings of Stevens and Campion's [88] study. In the online learning aptitude scale, the students had a sufficient ability in both (a) perceived technical skills of online learning and (b) comfort level with online learning environments, which was consistent with the findings of Xiong et al. [21].

In general, considering the age range of the participants in the present study, they can be described as a network generation, whose life and surroundings are surrounded by various technologies [89]. Therefore, it is expected that students will have high technological skills [74] and will be ready to adopt new technologies that are in line with today's technological developments. In the same line, Hung [90] reported that college students nowadays are very confident in the skills of computer/network (such as managing software, searching for information online, and perform the functions of the basic software), which is required for online learning. Thus, the results of this study also confirmed this and showed that higher education students were prepared to engage in CSCL environments, at least in the context of Iran.

It is worth mentioning that although the students at this age range may have more skills in the use of computers and electronic communication tools [23], they nevertheless need to have psychological readiness to participate in CSCL, the ability to conduct behaviors related to collaborative learning, and the ability to adapt to online learning environments. Therefore, policymakers and educational decision-makers at the university are expected to plan for CSCL by assessing the level of students' psychological readiness and ability in each period.

*5.2. Research Question 2*

There was a significant difference in the male students' learning online aptitude compared to the female students, suggesting that gender-specific behavioral patterns might differentiate women in the use of the CSCL environment [91]. According to Orji [92], there is a difference between men and women in a variety of areas, such as e-mail, data retrieval, e-learning, and electronic communication technologies. However, our finding was inconsistent with the previous research findings, showing that there was no difference in the application of technology between male and female students [82,93]. As mentioned before, the reason for this inconsistency could be attributed to the different cultural backgrounds, students' characteristics, and online learning environments [94–96].

Participants owning a personal computer, laptop, or tablet had a better performance on perceived readiness for CSCL scale and online learning aptitude compared to those who did not hold such electronic devices. In line with this finding, McCoy [74] and Nami and Vaezi [23] found that owning a personal computer could have a significant role in technology self-efficacy and overall technological skills. Therefore, this finding supported the hypothesis that computer ownership, and, in general, exposure to these technologies would increase the readiness of students for CSCL and the development of online learning skills. Hence, if course designers and decision-makers at the university intend to use the CSCL approach, they should provide access to such technologies for students.

The students' major of study was another important factor playing a role in their online learning aptitude. The results showed that the students in engineering and basic science groups had better performance in terms of online learning aptitude than students in the humanities group. In agreement with this finding, Sun and Rueda [24] reported that students at the faculty of Engineering had higher computer self-efficacy than those in other faculties. The findings of Nami and Vaezi [23] also indicated that the major of study had a significant role in the technology-related knowledge of the participants and that the students of computer engineering had a significant difference with the students of other disciplines. This difference could be attributed to previous student experiences in terms of computer use or to the greater interest of these students in technology. Especially in the case of the humanities

group, given the nature of the field, it is less probable for students to interact with technology and technological environments and devices in comparison with the engineering and basic science groups.

## 6. Limitations and Suggestions for Further Research

This study had some limitations that need to be considered. First, the results obtained in this study reflected the perception of a convenient sample of university students in four different academic fields with an unequal number and in the Iranian higher education context. Therefore, given the importance of contextual and cultural differences, for the sake of the generalizability of the findings and to avoid potential bias in the sampling method, future studies should be expanded with the use of a more diverse population of students from different universities, fields, and countries based on random sampling. Second, the data were collected only from the freshman students in the second semester and limited to the reflection of the students' perceived readiness for CSCL. Hence, to achieve a more comprehensive understanding of the readiness of students to implement such programs, the perception of junior and senior students, who are likely to spend more time working with computers and technology, should be studied. The reason is, students with more experience in accessing technology are more likely to be prepared to implement technology-related programs.

The theoretical framework of the present study was based on the framework provided by Xiong et al. [21] to assess students' perceived readiness in CSCL. The use of other models or frameworks may lead to different results. In addition, the use of other frameworks or models (such as Parasuraman's [57] technology readiness index) for assessing students' readiness in CSCL can increase the validity of the findings and add to the richness of the studies conducted in this area. Finally, this study focused only on three demographic variables (i.e., gender, major of study, and computer ownership), which are among the important variables that may have a role in students' readiness for a technology-based learning environment, such as CSCL. Future research can study the possible roles of other variables, such as age, previous experience, education, nationality, and cultural values [97,98], in students' readiness for CSCL.

**Author Contributions:** Conceptualization, G.K., O.N., and M.F.; Data curation, O.N. and M.F.; Formal analysis, G.K. and O.N.; Investigation, G.K.; Methodology, G.K. and O.N.; Project administration, G.K. and E.T.; Resources, G.K., M.F., and E.T.; Software, G.K.; Supervision, O.N. and E.T.; Validation, O.N.; Visualization, G.K.; Writing—original draft, G.K.; Writing—review and editing, O.N. and M.F. All authors have read and agreed to the published version of the manuscript.

**Funding:** This research received no external funding.

**Acknowledgments:** The authors would like to thank all the university students who voluntarily participated in this study.

**Conflicts of Interest:** The authors declare that there is no conflict of interest regarding the publication of this article.

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
