# Peer review of "Higher Education Students’ Perceived Readiness for Computer-Supported Collaborative Learning"

_mti, doi:10.3390/mti4020011_

Round 1

Reviewer 1 Report

Dear Authors,

In your paper, you have studied the perceived CSCL readiness of higher education students in Iran. Also, you examined the relation between some demographic information and the perceived CSCL readiness. You conducted a quantitative study by distributing a questionnaire developed based on the Xing instrument.

I read your article with much pleasure. It is a well-structure article and reads well. Furthermore, it sheds light on the CSCL readiness of Iranian higher education students and given the high uptake of higher education in Iran, the results of such studies can contribute to improving the quality of higher education in Iran.

I would suggest the following comments with regard to your submission:

-At line 69, there is a reference to the self-management skills of students. For a critical reader, the possible link between self-management and self-regulated learning skills might be a question.

-in sections 1.3.1 and 1.3.2, the relation between "gender difference" and "major of study" and the perceived readiness for CSCL in the light of the literature is illustrated. It seems that this is a highly context-dependent (it might differ from country to country) relation, for example, the relation between "gender difference" and the perceived readiness of CSCL. Accordingly, I would suggest mentioning the context of the conducted studies you have cited them in these sections.

-There are few typos at lines:  80 (ownerships), 102 (computer-scored), 113 (higher-education), 194 (owning own device)

-The instrument (Xing's instrument) used to collect the respondents' perception is missing. I would suggest to include it (or samples of its items) either in the appendix or in the text.

-In table 2, the meaning and calculation of "score range" needs more explanation.

Good luck!

Reviewer 2 Report

-Sections 1.1, 1.2, and 1.3 should be expanded to incorporate more of the literature. Moreover, if the authors wish to include a discussion of motivation, they should engage with prevalent theories. (The field of motivation is much larger than "extrinsic vs. intrinsic." 

-Both sections 1.1 and 1.2 rely too heavily on Xiong et al. 

-"There is a difference between men and women in terms of online learning behaviour. Men are more likely to use computers and new media for different purposes [25,44,45]." This is a bold and largely unsupported claim. The articles cited to support it are not generalizable to all men and women. Examining gender differences is fine, supposing that there are stark gender differences from the outset, with little evidence to support the claim, isn't. 

-RQ1 is unanswerable given the design, since the author's sample cannot generalize to the entire Iranian college student population. 

-The authors need to compute reliability for each of the sub-scales, not the scale as a whole. 

-Were computers, tablets, and laptops collapsed into one category? It would be good to see a more granular breakdown. (Tablets in particular are functionally different when compared to desktops and laptops.)

-How does having majority humanities majors affect your interpretation of results?

-Table 2 should just report means and standard deviations. Ranges aren't important. That said, it seems as though the authors summed the scores? Why not create a composite variable by taking the average of each sub scale? That is usually how this sort of analysis is performed. 

-The design is correlational; authors should remove language that is causal (e.g., affect)

-RQ1 is not addressed in the results section. Why is it included in the discussion? If RQ1 does not have analysis related to it, then it should be dropped entirely. 

-
